

# A Livestock Trampling Function for Potential Emission Rate of Wind-blown Dust in a Mongolian Temperate Grassland

Erdenebayar Munkhtsetseg[1], Masato Shinoda[2], Masahide Ishizuka[3], Masao Mikami[4], Reiji Kimura[5], and George Nikolich[6]

[1]School of Applied Sciences and Engineering, National University of Mongolia, Ulaanbaatar, Mongolia
[2]Graduate School of Environmental Studies, Nagoya University, Nagoya, Japan
[3]Kagawa University, Takamatsu, Japan
[4]Meteorological Research Institute, Tsukuba, Japan
[5]Arid Land Research Center, Tottori University, Tottori, Japan
[6]Desert Research Institute, Nevada University, Las Vegas, USA

*Correspondence to:* E.Munkhtsetseg (munkhtsetseg.e@seas.num.edu.mn)

**Abstract.** Mongolian Grasslands is one of the natural dust source regions and it contributes to anthropogenic dust due to its long tradition of raising livestock. Decades of abrupt changes in a nomadic society necessitate a study on effects of livestock trampling on dust emissions, so that research studies may help maintain sustainable ecosystem and well-conditioned atmospheric environment. For scaling the effect strength of trampling, therefore, we conducted a mini-wind tunnel experiment (by
PI-SWERL® device) to measure dust emission fluxes from trampling (at 3 disturbance levels of livestock density, $N$) and zero trampling (the background level) at test areas in a Mongolian temperate grassland. We found the substantial increase in dust emissions due to the livestock trampling. This positive effect of trampling on dust emissions was persistent throughout all wind friction velocities, $u_*$ (varying from 0.44 to 0.82 ms$^{-1}$). Significantly higher dust loading had occurred after a certain disturbance level has reached by the livestock trampling. Our result suggests that both friction velocity ($u_*$) and disturbance
level of livestock density ($N$) has enormous combinational effect on dust emission from trampling test surface. Furthermore, we successfully developed a livestock trampling function dependant on $u_*$ and $N$. In the livestock trampling function, almost 4 times greater of determinant power for $u_*$ than of for $N$ was determined ($f_L(N, u_*) \sim N^{1.1} u_*^4$). It points that the effect strength of trampling get magnified with an increase in $u_*$, and, therefore dust will emit much as stronger wind prevails at the trampled surfaces. This finding indicates that the effect of trampling can be seen or get into a play in emission when wind is
strong. It emphasizes that a better management for livestock allocation coupled with strategies to prevent dust loads is needed, however, there are many uncertainties and assumptions to be improved in this study. The applicability of our result is feasible with a care to other areas beyond the study location.

   Keywords: *surface disturbance; anthropogenic dust; PI-SWERL® mini wind tunnel; livestock density; Cobb-Douglas formula; friction velocity*



## 1 Introduction

Mongolian Grasslands is one of the natural dust source regions and it contributes to anthropogenic dust due to its long tradition of raising livestock. Mongolian ecosystem is generally sensitive to any external disturbance of the environment, natural or human, such as climate change or human activities (Peters, 2002; Pogue and Schnell, 2001). The projected increasing aridity

warns that enhanced warming (climate change) coupled with rapidly increasing human activities will further exacarbate the risk of land degradation and desertification in the near future in the drylands (Huang et al., 2016). Specifically, the major source regions of Asian dust has expanded from northwestern China to the Gobi Desert in Inner Mongolia (Wang et al., 2008; Fu et al., 2008). Livestock population has been increased substantially in the past decades (25 ml. in 1990, 30 ml. in 2000, 61 ml. in 2016) and it is projected to persist into the future (Shabb et al., 2013). Natural grassland exposures to livestock trampling,

overgrazing and road vehicle traffic are some of the most prevalent modifiable risk factors for dust emissions in Mongolia. Animal husbandry will contribute to atmospheric dust loading through degraded and disturbed land by i) grazing pressure and ii) livestock trampling (trampling pressure). In water and nutrient-limited environments, increased grazing is expected to increase plant mortality and ultimately decrease species richness (Huston and Huston, 1994; Proulx and Mazumder, 1998). The effect of grazing on land degradation that manifested with the declined plant species composition, low productivity and

poor soil fertility (Steffens et al., 2008; Li et al., 2009) is well established, and evidence that these outcomes are more severe in areas close to urban settlements and water resources (e.g., Mandakh et al., 2007; Wang et al., 2013; Hilker et al., 2014; Fujita et al., 2013; Bat-Oyun et al., 2016). This grazing pressure has been linked to increased number of dust events through elevated erodibility of land surface (Kurosaki et al., 2011) and altered areas in land cover types (Wang et al., 2008; Fu et al., 2008; Huang et al., 2015). Most studies assessed anthropogenic dust based on land cover data (Tegen et al., 2004; Huang et al.,

2014). So that dust loads from grazing pressure has been included in the total anthropogenic dust, semi/directly, however, there still remain large uncertainties (Tegen et al., 2004; Ginoux et al., 2012; Huang et al., 2014).

It has been long stated that dust models require land-surface parameterizations to improve their prediction performances (Shao, 2000; Marticorena and Bergametti, 1995; Uno et al., 2006; Lee et al., 2013). Underscoring effect of animal tramplings on the dust emission might be hindered research community to predicting dust precisely, particularly for intensive grazed dust

sources. Although we have known for some time that livestock trampling is also important factor in anthropogenic dust, our knowledge of the magnitude of the relation, both for total dust loads and for dust loads from anthropogenic activities, and the land disturbance of trampling pressure in terms of total atmospheric dust from grasslands is limited. Previous studies have shown associations between impacts of mechanical disturbance on soil particle bonds (Hoffmann et al., 2008; Steffens et al., 2008) and dust emission strength (Neuman et al., 2009; Houser and Nickling, 2001; Baddock et al., 2011; Macpherson et al.,

2008; Belnap and Gillette, 1997; Belnap et al., 2007) they revealed a common consequence of an increased dust emissions. However, areas of interest in such studies have been selected consistently to be the playa, where surface stability is considered to be a critical control in dust emission dynamics of dry lakes. Aside from physical crusts of playas, disturbance has also been investigated for the erodibility of biologically crusted desert soils with field wind tunnels (e.g., Belnap and Gillette, 1997; Belnap et al., 2007). In such experiments, the simulation of disturbance often involves an artificial agent, and although





both are effective in disrupting consolidated surfaces and offering straightforward replication, quantifying the effect of a natural process of disturbance should also be of significant interest for understanding wind erosion. Very few studies focused on natural disturbance effects such as livestock trampling for dust emissions which produced limited data (Houser and Nickling, 2001; Baddock et al., 2011; Macpherson et al., 2008). Scarce and inconsistent data prevents scientists to parameterize the disturbance

effects on dust emissions and to scale its relative contribution to the atmospheric dust. The lack of consistency is attributable to the limited number of studies, the limited range and variable categorization of land disturbance and dust flux among studies, and possibly real differences between the effects of land disturbance on the dust emissions from some land-surface parameters. Given above background, we aimed to reveal effect of trampling on dust emission processes, in a quantitave manner. For scaling the effect strength of trampling, therefore, we conducted a mini-wind tunnel experiment to measure dust emission fluxes from

trampling and zero trampling test areas. For achieving study aim, we purposed (1) to investigate effect of livestock trampling on dust emissions (2) to develop a livestock trampling function for emission rate, and (3) to examine the reliability of the trampling function and its applicability for assessing anthropogenic dust. Successfully introduced livestock trampling function will be useful to quantify an emission rate from trampling areas; and further, it might provide an insight into the possible assessment of anthropogenic dust for grazing areas. It should be mentioned that, our dust data represents the potential dust emission, as

a restriction of wind tunnel measurements. PI-SWERL® mini wind tunnel was successfully being used on playa surfaces to produce potential erodibility estimates (Etyemezian et al., 2007) that validated using conventional wind tunnel data Sweeney et al., 2008. This PI-SWERL® was also successfully used to investigate dust emission on surfaces in the Mongolian temperate steppe grassland (Munkhtsetseg et al., 2016).

## 2   Study materials

### 2.1   A study site description

Mongolian grasslands occupy over 80% of its total territory (equal to 113.1 ml. hectare). According to FAO 2010, as much as one-third of total pastures is under utilized. Most unused land is far from administrative centers and many herders are increasingly loath to travel that far, especially when infrastructure is deficient. Every year new wells operates, but huge number of wells still remains out of operation, resulting 10.7 ml. hectare of pasture that cannot be used because of lack of water (Suttie

et al., 2005). According to spatial density of livestock in Mongolia (Saizen et al., 2010), the largest number density of livestock is located on the Mongolian steppe grassland. The impact of grazing on plant diversity varies across environmental gradients of precipitation and soil fertility (Milchunas et al., 1988). In the desert-steppe zone, species richness was lower in the drier years but did not vary with grazing pressure. In the steppe zone, species richness varied significantly with grazing pressure but did not vary between years. species richness is not impacted by grazing gradient in desert steppe, but it is in the steppe (Cheng

et al., 2011). Consequently, the Mongolian steppe has been impacted the most by the grazing and trampling.

  Our study was carried out in Bayan-Unjuul (sum center) located in a temperate Mongolian steppe (Fig. 1a; 47°02′38.5″N, 105°56′55″E). Nomads and settlements of this sum have raised a large number of livestock, and they rank at number 30 out of 329 sums (Saizen et al., 2010). Last decade, number of dust events associated with wind erodibility has increased by 30%



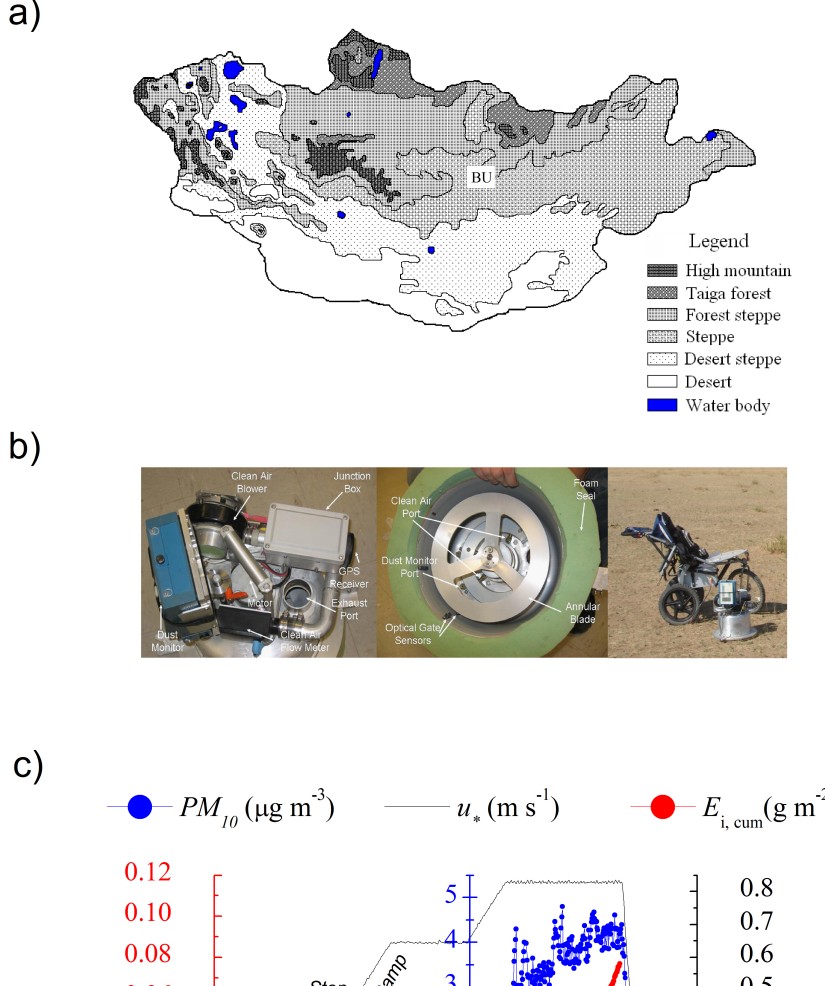

**Figure 1.** (a) BU (Bayan-Unjuul) denotes the location of the study site in with respect with to vegetation zones in Mongolia; (b) Pictorial illustrations of PI-SWERL®, top view at on the left, bottom view in the middle, and in the field situation at on the right sides; (c) An example data trace of $PM_{10}$ concentration and the cumulative dust emission ($E_{i,cum}$) associated with friction velocity ($u_*$) during PI-SWERL® measurement period (t).





in Bayan-Onjuul (Kurosaki et al., 2011). This is an area where dust emission activity has been monitored on a long-term basis (Shinoda et al., 2010a) at a dust observation site (DOS) adjacent to the study site (Fig. 1a). According to long-term meteorological observations made at the Institute of Meteorology and Hydrology of Mongolia's (IMH) monitoring station located near the site, the prevailing wind direction is northwest. Mean annual precipitation is 163 mm, and mean temperature is 0.1°c for

the period 1995 to 2005 (Shinoda et al., 2010b). Soil texture is dominated by sand (98.1% , with only 1.3% clay, and 0.6% silt (Table 1; Shinoda et al. (2010a)).

## 2.2 Wind tunnel experiment

### 2.2.1 PI-SWERL® mini wind tunnel

The PI-SWERL® consists of a computer-controlled 24-volt DC motor attached to the top of an open-bottomed cylindrical chamber 0.20 m high and 0.30 m in diameter (Figure 1b). Inside the chamber there is a flat annular ring (width = 0.06 m) with an outer diameter of 0.25 m, which is positioned 0.05 m above the soil surface (Figure 1b). As the annular ring revolves about its center axis, a velocity gradient forms between the flat bottom of the ring and the ground creating a shear stress $Nm^{-2}$ on the surface (Etyemezian et al., 2007). Dust and sand are mobilized by the shear stress generated by the

rotating ring. Dust concentration ($PM_{10}$) within the chamber that encloses the annular ring is measured by a nephelometer-style instrument, the 8520 DustTrak (TSI, Inc., Shoreview MN). The PI-SWERL® tests measure the potential fugitive PM10 dust emissions from the surface at different friction velocity $u_*$ ($ms^{-1}$ ) corresponding at the high end to a wind speed of approximately 30 $ms^{-1}$ at 2 m above ground level (AGL). In this experiment, the rotation per minutespeed RPM (in rpm) of the annular ring was converted correlated to with a corresponding (in $ms^{-1}$ ) friction velocity. The measured data by the

PI-SWERL® instrument were analyzed using the miniature PI-SWERL® user's manual (version 4.2) (DUST-QUANT, 2009). Each PI-SWERL® experiment consisted of friction velocities vary from 0.16 to 0.82 $ms^{-1}$ . Depending on the different friction velocity, six levels are identified (i = 1, 6) within each PI-SWERL experiment. Four levels include two gradual increases in $u_*$ 0.54, 0.73 $ms^{-1}$ (ramp properties) separated by three constant $u_*$ settings of 0.44, 0.64, and 0.82 $ms^{-1}$ (step properties) dust emission flux was used (Fig. 2c). When performing the dust measurements by PI-SWERL® , we avoided duplicating

measurements on the same location by shifting its position each time.

### 2.2.2 Experimental area setting

While grazing, livestock leaves behind its trampling trace; therefore, we schemed a trampling route based on grazing route (Fig.2a). Many studies proved that livestock density (i.e.,grazing pressure) is usually highest close to water sources or settlements and decreases with distance away from such localities (ANDREW and LANGE, 1986; Fernandez-Gimenez and Allen-

Diaz, 2001; Landsberg et al., 2003; Sasaki et al., 2008; Cheng et al., 2011). According to (Stumpp et al., 2005) the livestock spatial densities were higher in the first 300 m of the transects from the local centers. This finding of the heavy grazing with a 'radial gradient' was also found at our study site (Cheng et al., 2011), which spots a trampling-active area. The trampling-active



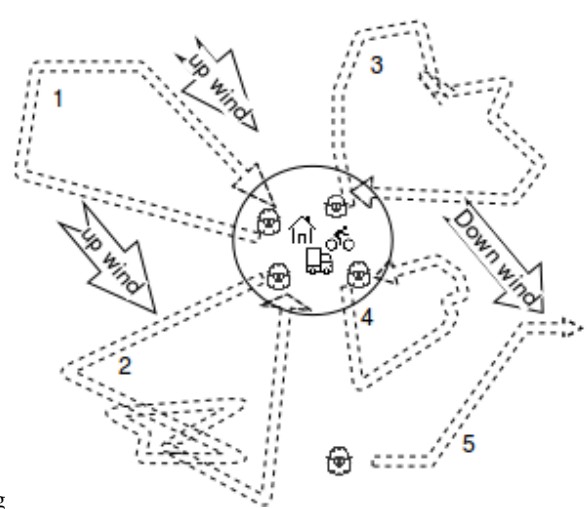

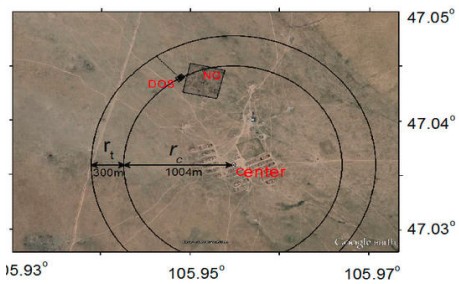

2a.png

(a) A schematic drawing of grazing route around administrative center (or well). Livestock will graze on daily routine depending on weather and fodder source. Type I route (marked by 1, 2, and 3) usually happens good weather condition with rich fodder; Type II route (marked by 4) happens during bad weather condition, like spring and winter; Type III route (marked by 5) is called *otor*.

(b) Annulus area selected for this study. $r_c$ is the distance from sum center (*Center*) to the inner circle for the selected annulus are and $r_t$ is the width of annulus area

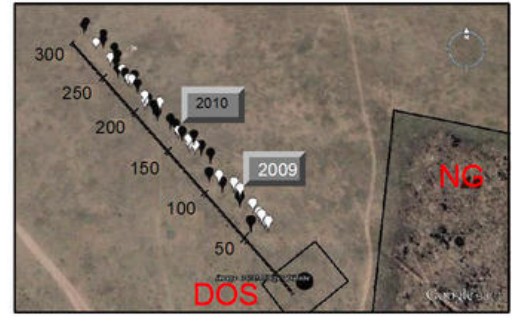

(c) PI-SWERL® experimental test areas (transect lines; and Dust Observation Site (*DOS*)). White and dark balloons presents dust sampling points along the transect in 2009 and 2010.

**Figure 2.** Experimental settings

area (with 300 m transect) close to local centers is reasonable from the view point of livestock trampling routes as well. Three types of pictorial livestock trampling routes could be illustrated based on published result (Suttie et al., 2005) on a seasonal and spatial variability of trampling density in reference to grazing habits in seasons and animal types (Fig.2a). Type I, a long grazing route, draws summer and autumn pasture is usually grazed in common, with few problems of access or dispute. Type

II, a short grazing route, draws the winter and spring camps and grazing are the key to the herders overall system; (at a season when feed is very scarce) each must provide shelter as well as accessible forage through that difficult season (Fig.2a). Type III, a distanced grazing route, draws taking livestock to more distant fattening pastures (*otor*) is an important part of well organized herding and, if done with skill, can greatly improve the condition of stock before the long winter. Horses and cattle may be left to graze, except those being milked. So, measuring dust emissions at the area close to the local center will reflect on the

trampling activity.

  Our study aimed to measure dust emission affected only by livestock trampling versus no-trampling. Therefore, we focused





to do PI-SWERL® mini wind tunnel experiment under similar weather and surface aeolian conditions at the trampled and no-trampled areas. Performing PI-SWERL® mini wind experiment for a short period of time will enable us to avoid weather changes. Experimental test area of livestock trampling was selected to be close to the no-trampling area where both areas are
subjected to similar surface aeolian condition. Hence, at foremost, trampling active area at our site was presented by annulus area enclosed by inner and outer circles (Fig.2b). Inner circle excludes a residential area where land is disturbed mainly by local people's daily activities, while outer circle delimits trampling activity of 300 m from the local (outer) center. The residential area was defined with a radius starting from the BU sum center to the most distanced object. It is well known that sand and dust particles transported by wind likely to deposits on downwind lee, when distracted by rough objects like vegetation, shelter areas
or buildings. This condition results a distinct fractions of sand and dust on land surface, which will produce a differenial dust emissions. As mentioned above, the prevailing wind direction (*NW* at our site) will differentiate potent emission into upwind and downwind areas. In order to avoid or reduce a possible source of data uncertainty by of the aeolian processes at the site, we narrowed our area of interest into the upwind area of the trampling active area. Further, regarding all possible requirements, the transect line shown in Figure 2c was mandatory to run PI-SWERL experiment.

### 2.2.3  Experimental conditions of land surface

**Analogous conditions of vegetation and soil moisture**

The vegetation and pebble survey was defined along the transect distanced at 50, 150, 200, and 300 m away from the DOS within a 1 m × 1 m plot (Table 1) (Munkhtsetseg et al., 2016). In the two springs of 2009 and 2010, vegetation conditions were similar. Vegetation covers were 2.4 and 2.3 percentages during the measurement periods in 2009 and 2010, respectively.
These seasonal conditions resulted in sparse vegetation growth and exposed large portions of the surface area to be free of vegetation. This open area enabled us to run PI-SWERL® wind tunnel which has a limitation in- measuring dust emissions over a vegetated area where vegetation height is above 4cm. Sparse vegetation growth during the measurement periods and a small size of PI-SWERL® (an effective area of 0.026 m$^2$) enabled us plenty of bare surfaces to conduct PI-SWERL® measurements. Therefore, our dust measurements by PI-SWERL® were not influenced by vegetation roughness. Recent study revealed that soil moisture has a clear seasonal variation in Mongolia with the lowest value in the spring times (Nandintsetseg and Shinoda, 2015). Consequently, the spring is recognized as a dust favorable season due to its low seasonal precipitation (Shinoda et al., 2011). Averaged soil moisture values were 0.0022 and 0.0024 gg$^{-1}$ in 2009 and 2010, respectively. Soil moisture values showed a subtle change in standard deviations of soil moisture. Consequently, these standard deviations revealed insignificant
5 changes in soil moisture among the transect lines for each year. As a temporal variation between the 2-year study period, the difference in averaged soil moisture values on these two springs was equal to 0.0002 gg$^{-1}$, which is insignificant amount that can alter amount of dust emissions (Fécan et al., 1998). These climatic conditions and above mentioned experimental settings clearly indicate that both soil moisture and vegetation conditions were not influential factors in altering dust emissions from bare, non-trampled and systematically trampled surfaces in 2010; and the naturally trampled surfaces in 2009 and 2010.



**Table 1.** Land surface and soil size characteristics in the study area

| | Pebble cover (%) | Soil texture fraction[1] (10 cm depth) (%) | | | Vegetation cover[2] (%) | | Soil moisture[2] ($gg^{-1}$) | |
|---|---|---|---|---|---|---|---|---|
| | (<30 mm) | Clay (<0.002 mm) | Silt (0.002-0.02 mm) | Sand (0.02-2 mm) | in 2009 | in 2010 | in 2009 | in 2010 |
| Mean | 2.4 | 1.3 | 0.6 | 98.1 | 2.4 | 2.3 | 0.0022 | 0.0024 |
| SD | 0.18 | 0.1 | 0.1 | 0.1 | 0.3 | 0.2 | 0.0001 | 0.00012 |

[1] is defined in Shinoda et al. (2010a); [2] is presented in Munkhtsetseg et al. (2016)

### Livestock trampling density

The total numbers of livestock at bag-scale for Bayan-Onjuul subdistrict were counted as 52378 and 43709 for 2009 and 2010, respectively (National statistical Organization reports, 2009; 2010). We calculated livestock densities in the annulus area (Fig. 2b) for a given year, as presented in Eq.(1):

$$N = \frac{10^4 num}{\pi(r_c + r_t)^2 - \pi r_c^2} \tag{1}$$

where $N$ is livestock density in head per hectare ($Head\,ha^{-1}$); *num* is total livestock in a head; $r_c$ (=1004) is the radius distance from the center to the transect start-line in meter; $r_t$ (=300) is the transect line in meter; $10^4$ is a unit conversion of square meter to hectare (Fig. 2c). Total livestock in a head is the total number of 5 animals: sheep, goat, camel, cattle, and horse that are traditionally herded by the nomads. The calculated livestock densities were 241 and 201 $Head\,ha^{-1}$ along transect lines in 2009 and 2010, respectively.

As for trampling inside DOS fenced area, a calculation of livestock density was followed a basic procedure. A total fenced area of DOS was 50 m x 35 m. Inside DOS fence, sheep movement was constrained into a subarea of 8 m x 35 m to ensure that allocated meteorological equipments would not be damaged. Livestock density inside DOS, therefore, calculated as a spatial distribution total sheep to the enclosed area of 8 m x 35 m, and it estimated as 250 $Head\,ha^{-1}$.

We assumed that all types of livestock (small and large rumnitants) has the same effect on land surface trampling, irrespective of the size or distribution of the footprints. In addition, we made no distinction between the weights of the different livestock species. However, the potential variability due to the difference in weights warrants further investigation. (Xu, 2014) tested the quantity of dust emitted from vehicles and found that it varied with the weight of the vehicles.

### 2.2.4 Field experiment

Figure 3 presents experimental details including experimental plots, measurement replications and associated livestock density ($N$). Inside the DOS, where is no-trampling area ($N$=0), we collected 7 replicative dust data on 16 May, 2010. At the same day, we collected 4 replicative dust data after 5 hours of grazing of 7 sheep ($N_{250}$), those herded into inside the DOS (Figure 3a). We collected 21 replicative dust data along the naturally trampled transect line (shown Fig. 2c) with $N$ of 241 $Head\,ha^{-1}$ ($N_{241}$)on



15 May 2009. On following winter, livestock denity at our study site was reduced due to the moderate dzud (Mongolian word indicating harsh winter conditions contributes to livestock mortality) (Natsagdorj and Dulamsuren, 2001; Begzsuren et al., 2004). We collected 25 replicative dust data along the naturally trampled transect line with $N$ of 201 Headha$^{-1}$ ($N_{201}$) on 15 May, 2010 (Fig. 3b).

All dust emission data was obtained by the PI-SWERL® mini mind tunnel. For producing replicatve data, we avoided to run PI-SWERL experiment on the same spot by shifting to the other area. Additionally, we tried to perform all PI-SWERL® measurements at the same day to obtain unbiased data by weather changes from day to day. Since April 2008, DOS was fenced to keep out livestock; no any livestock trampling for a 2 year-period. These measured dust fluxes, on bare surfaces inside DOS (fenced to keep livestock out) was considered as a reference dust for non-trampled surfaces ($F_{REF}$).

Moreover, livestock trampling intensities for all 3 types of measurements was likely subject to on annual basis. Because, dust

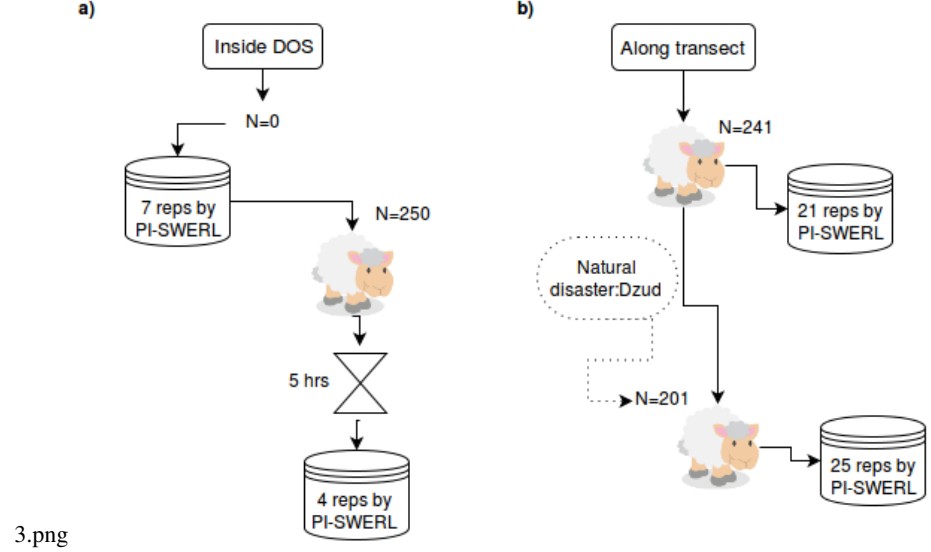

**Figure 3.** A schematic flowchart of PI-SWERL® experimental test a) for zero and trampling of $N_{250}$ inside *DOS*; and b) for trampling of $N_{201}$ and $N_{241}$ along transect areas. PI-SWERL® experimental replications for each dataset is marked as *reps*.

emissions at the naturally trampled transect were measured on annual basis (at the springs 2009 and 2010). As for dust for N=250 Headha$^{-1}$, 5 hour of grazing is also annual if we considere that average walking speed for livestock is 314 mh$^{-1}$ (equal to approximately within 11.5 s time-period covers a 1 m path) (Plachter and Hampicke, 2010). Assuming that the livestock pass 4 times (from sum center to grazing area and vise versa) along the transect lines of the ring area in a day, this

will resulted in a yield of 1460 times passages per a year. On annual basis, livestock walk over and over a 1 meter path for a time period of 4.6 h (11.5 s × 1460 times=16790 s). This finding can be used to estimate an average time period of livestock trampling in the fence. Due to limited space, the livestock inside the fence was in a near static movement by not walking the path. This condition enabled us to assume that each sheep stands in the static state covering around 1 m path with respect





to their body size. Thereafter livestock trampling continued for a half day ($\approx 5$ h) on bare surfaces inside DOS, after which
systematic trampled dust emissions measurements were conducted by the PI-SWERL® instrument.

## 3   Study methods

### 3.1   Anthropogenic dust induced by trampling of livestock

In this study, anthropogenic dust emission flux, $F_N$, is considered two main assumptions. These two main assumptions are: 1)
dust amount will be larger at the livestock trampling test areas than at the zero trampling surfaces ($F_N \geq F_{REF}$); 2) increased
dust amount will be subject to a trampling function, $f_L(N, u_*)$. Therefore, our dust flux formula was expressed as shown in
Eq (2):

$$F_N = F_{REF}(1 + f_L(N, u_*)) \tag{2}$$

where $F_N$ and $F_{REF}$ are dust emissions from test areas of trampling and zero-trampling, respectively ($\mu g m^{-2} s^{-1}$).
In equation (2), $F_{REF}$ is included for a quantity term only, whereas $f_L(N, u_*)$ is referred to a physical term for defining $F_N$.

### 3.2   A formula of livestock trampling function for dust emission

On a physical basis, livestock trampling weakens soil particle bonds to result an ease dust inputs released by wind blows ($u_*$)
into the atmosphere (Baddock et al., 2011; Macpherson et al., 2008). Surface disturbance does not directly cause dust emission
but it does recover surface available dust (Zhang et al., 2016). It suggests that a formula of livestock trampling function
$f_L(N, u_*)$ should be deriven by livestock density (effect of trampling) and $u_*$ (wind blown dust). Moreover, $f_L(N, u_*)$ should
be a dust product (of $N$ and $u_*$) to reflect the physical term of $F_N$. Thus, we employed the Cobb-Douglas function for defining
our livestock trampling formula.
The Cobb-Douglas function is widely used in economics to show the relationship between input factors and the level of total
production, *Y*, as presented in Eq.(3):

$$Y = AL^\beta K^\alpha \tag{3}$$

where, *A* is total factor productivity, *L* is labor input, *K* is capital input; $\alpha$ and $\beta$ are the measures the responsiveness of output
to a change in levels of either *L* or *K* used in production, *Y*. If $\alpha + \beta = 1$, the production function has constant returns to scale,
meaning that doubling the usage of capital K and labor L will also double output Y. If $\alpha + \beta < 1$, returns to scale are decreasing,
and if $\alpha + \beta > 1$, returns to scale are increasing.
In last decades, this function has been widely used in environmental studies, such as, to show the relationship between human
population as a capital input and pollutant emissions as a labor inputs, to output activities to economic growth etc (Labini,
1995; Dong et al., 2013; Cheng et al., 2015). Similarly to this, we employed Cobb-Douglass function to define a formula of
$f_L(N, u_*)$ as a product of $N$ and $u_*$ considering trampling increases the weak-bonded dust particles by a labor of livestock





density, (as $L$, labor), and these dust particles are carried away by wind, $u_*$ (as $K$, capital). If we replace our inputs in eq.(3),
the formula of $f_L(N, u_*)$ will be defined as eq.(4):

$$f_L(N, u_*) = A N^\beta u_*^\alpha \qquad (4)$$

where, $N$ is livestock density ($\mathrm{headha}^{-1}$) and $u_*$ is friction velocity ($\mathrm{ms}^{-1}$).

## 4   Data analysis

It was demonstated that microscale disturbance by marmots creates spatially heterogeneous grasslands at a fine scale (Yoshihara
et al., 2010). Moreover, it was emphasized that the livestock modified spatial heterogeneity at the landscape scale, whereas
marmots modified spatial heterogeneity at the local scale (Yoshihara et al., 2010). Thus, in this study, we used the means of
dust emission for investigating effect of trampling.

We calculated the mean values of dust emissions by averaging measured dust fluxes for each livestock density groups ($N$ of
0, 201, 241 and 250 $\mathrm{Headha}^{-1}$ ). Data from each group for each friction velocity were treated separately. We tested datasets
normality with Shapiro-Wilk test. The Shapiro-Wilk test is widely used to define the normality when the sample number is
below 50. It is believed that it works well with samples from 4 to 2000 (Razali et al., 2011). One-way analysis of variance
(ANOVA) was used to determine if there is a difference in the mean dust emissions of livestock trampled surfaces (with
livestock densities of 201, 241 and 250 $\mathrm{Headha}^{-1}$ ) from zero trampled surface. The determined coefficients for A, $\beta$ and $\alpha$
are equal to 0.06853, 1.1 and 4 (Fig. 3c). We used the least square optimization method with Levenberg-Marquardt algorithm
(Moré, 1978) to determine the coefficients of A, $\beta$ and $\alpha$. The Levenberg-Marquardt (LM) algorithm is an iterative technique
that locates the minimum of a function that is expressed as the sum of squares of nonlinear functions (Marquardt, 1963;
Lampton, 1997). It has become a standard technique for nonlinear least-squares problems and widely adopted in a broad
spectrum of disciplines. We employed OriginPro 8.1 Academic software (Northampton, MA 01060 USA) for calculating
statistics and determining the coefficients by the least square optimization method.

## 5   Results

### 10   5.1   Livestock trampling effects on dust emission

The mean rate of $PM_{10}$ emission from the test surface areas for each friction velocity of PI-SWERL® experiment reveals
greater detail concerning the behaviour of dust emission and the effect of trampling (Fig. 4)(Appendix 1). The dust emission
from the undisturbed, zero trampling surface at friction velocity $u_*$ of 0.44 $\mathrm{ms}^{-1}$ was low (10.5 $\mu g m^{-2} s^{-1}$). This was ele-
vated to 15.7 $\mu g m^{-2} s^{-1}$ at $u_*$ of 0.54 $\mathrm{ms}^{-1}$, and then backed to background level 10.1 $\mu g m^{-2} s^{-1}$ at 0.64 $\mathrm{ms}^{-1}$. A noticeably
increased emission rates of 39 and 37.3 $\mu g m^{-2} s^{-1}$ are seen at the $u_*$ of 0.73 and 0.82 $\mathrm{ms}^{-1}$ respectively, however, their dif-
ference was negligible. These dust emission behaviors in a change with $u_*$, which are in a sequential order for each PI-SWERL
experiment, suggest that our sandy soil of temperate grassland is somewhat similar to a supply-limited surface with successive




emission (Macpherson et al., 2008). In contrast, dust emission from trampling test areas presents that the disturbed, trampling surface is an unlimited-supply dust surface concerning its apparent increased emission rate with an increase in $u_*$ (Fig. 4),

except the case of $0.64\,\mathrm{ms}^{-1}$ subtle declined to $0.64\,\mathrm{ms}^{-1}$.

At friction velocity of $0.44\ \mathrm{ms}^{-1}$, although dust emission was almost doubled between zero trampling and $N_{250}$ trampling, this difference was not statistically significant (Fig. 4b). Additionally to this, trampling effect is visible when considering an increase in mean dust fluxes with all trampling densities of 201, 241 and 250 $\mathrm{head\ ha}^{-1}$ (Fig. 4b). However, a such increase is invalid if include dust flux at zero trampling in comparisons with those $N_{201}$ and $N_{241}$ tramplings, but their differences are

very small. We used Shapiro-Wilk test (with a significance of $\alpha = 0.05$) and standard deviation to assess whether the variables had a normal distribution and equilibrium or diverse variances in the statistical populations, respectively. Dust flux for zero trampling surface shows statistically significant with the normality. Contrastingly, the insignificant normalities is demonstrated with the trampled area datasets (Fig. 4a) along with larger standard deviations (Fig. 4b), those are resulted by scattered data points from their sample populations (see Fig. 4a; data points with box chart of $25^{th}$ and $75^{th}$ percentiles). Higher diversity of

dust fluxes presents morphological disparity and sedimentological diversification presence in livestock trampled test areas.

At moderate friction velocities of 0.54 and $0.64\ \mathrm{ms}^{-1}$, emission rates at $N_{250}$ trampling area was almost 5 times larger of that zero trampling, and their differences was statistically significant (One-way ANOVA test; p value with 0.05) (Fig. 4b, denoted by $*$). Trampling effect, which was visible for $u_*$ of $0.44\ \mathrm{ms}^{-1}$, is apparent when observing increases in mean dust fluxes with all trampling densities for $u_*$ of $0.54\ \mathrm{ms}^{-1}$, and for $0.64\ \mathrm{ms}^{-1}$ includes even non-trampling (Fig. 4b). The insignificant normalities of emission rates with trampling densities of $N_{201}$ and $N_{241}$ (Fig. 4a) along with larger standard deviations (Fig. 4b) are demonstrated, as it was also seen for $u_*$ of $0.44\ \mathrm{ms}^{-1}$. Emission rates with trampling densities of zero and $N_{250}$ presents significant normalities, and this significancy supports the difference of dust fluxes between zero trampling and $N_{250}$

trampling (Fig. 4b, denoted by $*$). Dust emission produced at $0.54\,\mathrm{ms}^{-1}$ was smaller than those at $0.64\,\mathrm{ms}^{-1}$ reflects similar surface emission characteristics to the undisturbed surface and types those are discussed by Macperson et al. 2008.

At high friction velocities of 0.73 and $0.82\ \mathrm{ms}^{-1}$, trampling effect is strongly pronounced. It can be seen in enlarged emission rates at all trampling area from that zero trampling; specifically, 5-10 times for $u_*$ of $0.73\ \mathrm{ms}^{-1}$, and 10-20 times for $u_*$ of $0.82\ \mathrm{ms}^{-1}$. Consequently, emission rates at $N_{201}$ and $N_{250}$ significantly differ from that zero trampling, which is supported

statitically by their significant normalities (Fig. 4b, denoted by $^*$). Moreover, an increase in mean dust fluxes with increase in $N$ for all trampling densities (including non-trampling) also perceives the effect of trampling.

Overall, the effect of trampling on dust emissions was persistent throughout all friction velocities. Significantly higher dust loading was occurred after a disturbance level has reached by the trampling $N_{250}$. But, the disturbance level was lowered with an increase in wind force, $u_*$. It indicates the effect of trampling can be seen or get into a play in emission as strong as dust

storm happens.



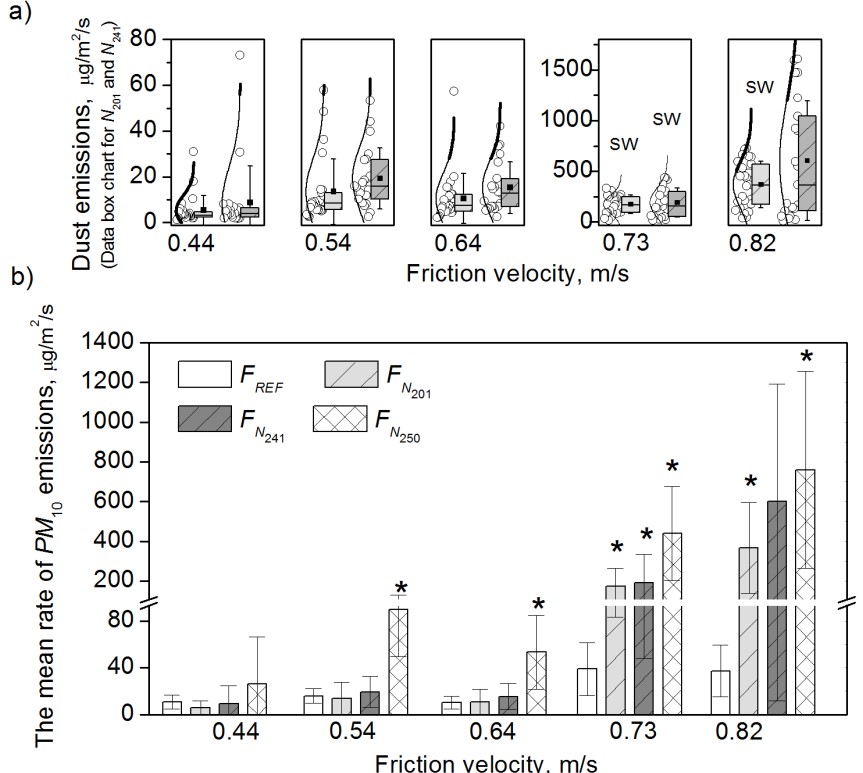

**Figure 4.** a) Measured dust fluxes from the trampled surfaces with $N$ of 201 and 241 Headha$^{-1}$. Open circles (∘) and curved lines (⎰) denote collected dust data and normal distributions. Center dots (.) and dashes (−) in the boxes denote means and medians of dust emissions. Opening and closing of the boxes presents 25 and $75^{th}$ percentiles for each dataset. *SW* denote statistical significant datasets with Shapiro-Wilk normality test. b) Mean dust emission fluxes with standard deviations on correspondent friction velocities. The significant differences (*p* value with the significant level of 0.05) for mean dust emissions of the trampled surfaces from the $F_{REF}$ on each friction velocity is denoted by ∗.

## 5.2 A livestock trampling function for dust emissions

The mean ratio of $F_N$ and $F_{REF}$ demonstrates the effect of trampling in dust emission as a relative rate scaling by $F_{REF}$ for each $u_*$. Because every mean ratio is calculated under the same shear force ($u_*$), it would remove the effect of $u_*$ in the relative rate. For a given trampling density, therefore, we expected a constancy in the relative rate among all $u_*$. But in reality, the mean ratio varies at the same trampling density (Fig. 5a; As for an example, $F_{N_{250}}/F_{REF}$ ranged from 2.5 to 20). This indicates that the effect of trampling can be dissimilar throughout $u_*$. Very interestingly, we found that the variabilities in the mean ratio is subject to changes in $u_*$ demonstrating positive relationships for all trampling test areas (Fig. 5a). Thus the effect strength of trampling is likely to be determined by $u_*$. Some of the mean ratios are less than 1, particularly at low friction velocities for livestock trampling with $N_{201}$ and $N_{241}$ Headha$^{-1}$, which also supports our idea on importance of high $u_*$ for revealing the





effect of trampling.

Beside of $u_*$, trampling density ($N$) also contributes the mean ratio variabilities to increase. It can be seen in the elevated mean ratios, those are increased from 10 times (at $N_{201}$) to 16 and 20 times (at $N_{241}$ and $N_{250}$, respectively), demonstrated for $u_*$ of 0.82 ms$^{-1}$ (Fig. 5b). However, the differences between $F_{N_{201}}/F_{REF}$ and $F_{N_{241}}/F_{REF}$ become smaller as the value in $u_*$

of 0.82 ms$^{-1}$ shifts to moderate and low values (Fig. 5b). This is consistent with the finding in section 5.1, that significantly higher dust loading occurs after a disturbance level has reached by the trampling $N_{250}$. But measuring the effect of trampling, in an interactive term of $u_*$ and $N$ is somewhat challenging.

Anthropogenic dust emission formula is defined in (2) (Section 3.1). If we divide both side of Equation (2) by $F_{REF}$, it will

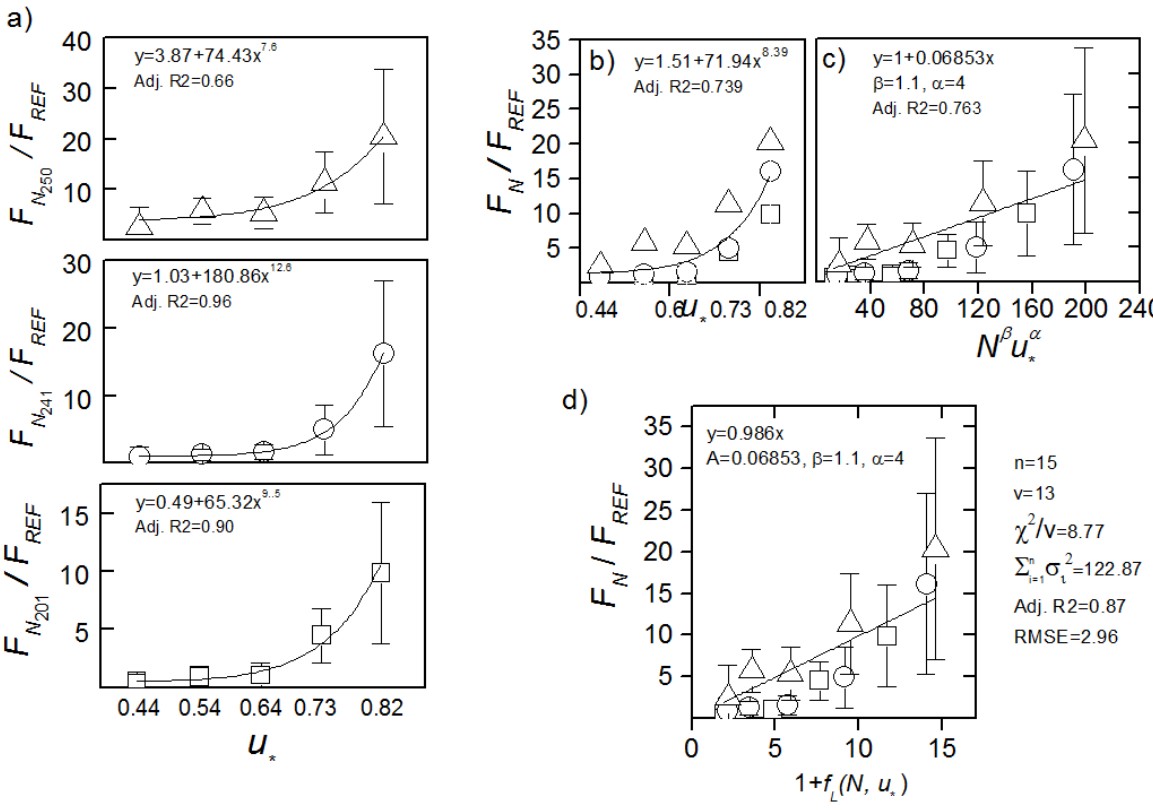

**Figure 5.** Dust emission ratio dependencies on friction velocity and livestock density. Open triangles ($\triangle$), circles ($\circ$), and squares ($\square$) denote dust emission values from the trampled surfaces with livestock densities of 250, 241, 201 Headha$^{-1}$, respectively. a) Relationships between the mean emission ratio (of $F_N/F_{REF}$) and $u_*$ for $N$ of 201, 241and 250 Headha$^{-1}$; b) A fitted relationship between dust emission ratios and $u_*$ for all livestock trampled surfaces; c) A relationship between dust emission ratio and Cobb-Douglas production function of $N$ and $u_*$; d) A plot of $F_N/F_{REF}$ versus $1 + f_L(N, u_*)$. $n$ denotes sample number, $v$ is degree of freedom, $\chi/v$ is the reduced chi square, $RMSE$ is root mean square error, Adj.$R^2$ is adjusted $r^2$, residual sum of squares.





yield an Equation (5). This equation suggests that the livestock trampling function $, f_L(N, u_*)$, can be obtained by the mean emission ratio, $F_N/F_{REF}$, presented in Figure 5b.

$$F_N/F_{REF} = 1 + f_L(N, u_*) \tag{5}$$

$$f_L(N, u_*) = 0.06853 N^{1.1} u_*^4, \quad \text{for } 240 < N \le 250 \text{ and } 0.44 \le u_* \le 0.82 \tag{6}$$

The determined A, $\alpha$ and $\beta$ for $f_L(N, u_*)$ illustrated in Figure 5c, and the defined $f_L(N, u_*)$ presented in Eq (6). Both of

$\alpha > 1$ and $\beta > 1$ indicates the increasing scale for each component return. This means that 15% of increase will return 16.6% in $N$ and 74.9% in $u_*$ of increased $f_L(N, u_*)$ output. Consequently, effect of trampling will be magnified by an increase in both $N$ and $u_*$. The magnitude strength is greater with a change in $u_*$ than in $N$.

We examined a performance of the defined $f_L(N, u_*)$, Eq.(6), plotting against $F_N/F_{REF}$ in Figure 5c. It validates the well-parameterization for $f_L(N, u_*)$, illustrating a reasonable error ($RMSE = 2.96$), a good fit ($Adj.r^2 = 0.87$) and a significant

value (p<0.05; by reduced $\chi^2$ test) between the term of $1 + f_L(N, u_*)$ and $F_N/F_{REF}$ (Fig. 5d). This ensures that livestock trampling function is applicable to assess the effect of trampling; however, a valid range is limited (Eq.(6)).

## 5.3 An application of $f_L(N, u_*)$ for assessing dust emission flux

Applying the results tackled in Section 5.2, anthropogenic dust can be assessed by Equation (7), however the valid range for $N$ is much narrow. This limits the usage of the Eq.(7) to assess $F_N$ in a broad range of areas (for Mongolia), where livestock

density varies spatially (Saizen et al., 2010).

$$F_N = F_{REF}(1 + f_L(N, u_*)), \quad \text{for } 200 \le N \le 250 \text{ and } 0.44 \le u_* \le 0.82 \tag{7}$$

To extend applicability of the eq.(7), we merged the valid range of $200 \le N \le 250$ down to $0 \le N \le 250$, concerning that equation (7) yields $F_N = F_{REF}$, when $N = 0$. This will also provide an opportunity to assess natural dust within the equation. Thus, we generalized eq.(7) into eq.(8), and for simplicity, we substituted $c u_*^4$ instead of $F_{REF}$ in it. Because it has been found

that dust flux is proportional to $u_*^4$ for a natural sandy soil (Zhang et al., 2016) in agreement with Gillette and Passi (1988) and in a range of power for $u_*$, ($u_*^{2.9} \sim u_*^{4.4}$), suggested by Shao (2008).

$$F = c u_*^4 (1 + f_L(N, u_*)), \quad \text{for } 0 \le N \le 250 \text{ and } 0.44 \le u_* \le 0.82 \tag{8}$$

where $F$ is dust emission from trampling and zero trampling test areas, $c$ is coefficient equal to 95.985. This dust flux formula is validated illustrating a reasonable error ($RMSE = 68.6$), a good fit ($Adj.r^2 = 0.93$) and a significant value (p<0.05; by

reduced $\chi^2$ test) between the modeled $F$ (eq.(8)) and measured $F$ in Figure 6. Moreover, the uncertainty of $F$ for eq.(8) is estimated to assess its error range (see Appendix 3). The error range is determined as equal to $\pm 0.36F$, and it reveals a reliable range of $F$ lies in $F = F \pm 0.36F$. These facts confirm that eq.(8) has a feasible potential to assess dust emission fluxes, properly. However, the threshold friction velocity should be included further for a better prediction performance, precisely (Gillette and Passi, 1988; Macpherson et al., 2008).





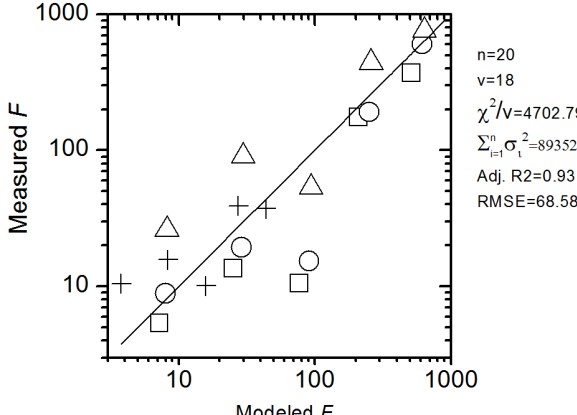

**Figure 6.** Modeled dust emission values versus measured dust emissions (in $log_{10}$ scale). Open triangles ($\triangle$), circles ($\circ$), and squares ($\square$) denote dust emission values from the trampling of $N$ of 250, 241, 201 $\text{Headha}^{-1}$, respectively. Plus symbols ($+$) denotes dust emission values from the zero trampling surfaces.

## 6 Discussion

### 6.1 The effect of trampling

We found substantial effect of the trampling on dust emission. The mean rate of $PM_{10}$ emission from the test surface areas for each friction velocity of PI-SWERL® experiment reveals greater detail concerning the behaviour of dust emission and the effect of trampling (Fig. 4).

The dust emission from the undisturbed, zero trampling surface at friction velocity $u_*$ of $0.44\,\text{ms}^{-1}$ was low ($10.5\,\mu gm^{-2}s^{-1}$). It was elevated to $15.7\,\mu gm^{-2}s^{-1}$ at $u_*$ of $0.54\,\text{ms}^{-1}$, and then backed to background level $10.1\,\mu gm^{-2}s^{-1}$ at $0.64\,\text{ms}^{-1}$. A noticeably increased emission rates of 39 and $37.3\,\mu gm^{-2}s^{-1}$ are seen at the $u_*$ of 0.73 and $0.82\,\text{ms}^{-1}$ respectively, however, their difference was negligible. These dust emission changeable behaviors in a change with $u_*$, those are in a sequential order of shear stress for each PI-SWERL experiment, suggest that our sandy soil of temperate grassland is somewhat similar to a supply-limited surface with successive emission (Macpherson et al., 2008). This is consistent with the hypothesis for supply-limited surfaces that the quantity of dust ejected into the atmosphere is controlled by the capacity of the surface to release fine particles(Nickling and Gillies, 1993).

In contrast to the undisturbed surface, that the disturbed, trampling surface behaves as an unlimited-supply dust surface, concerning its consistent increase in emission rate with an increase in $u_*$ (Fig. 4), except the case of $0.64\text{ms}^{-1}$ a subtle decline to $0.54\text{ms}^{-1}$. This shift in natural soil, from suply limitedness to unlimited supply surface, could be explained by the weakening of inter particle bonds, as a consequence of trampling (Belnap et al., 2007; Baddock et al., 2011; Macpherson et al., 2008). In some crusted desert soils with higher sand contents, disturbance can lead to increased sand availability and the occurrence of effective abrasion (e.g., Belnap and Gillette, 1997). In conjunction to this explanation, we observed increased dust emission



from the trampling test areas in comparison to those from zero trampling, despite similar ranges in shear velocity. Their differences was statistically significant (One-way ANOVA test; p value with 0.05) (Fig. 4b, denoted by ∗) at most of $u_*$, particularly for $N_{250}$ trampling. Observed emission rate at $N_{250}$ trampling were 26.1 and 760 $\mu g m^{-2} s^{-1}$, those value are approximately 2 and 20 times greater than those zero trampling, measured at $u_*$ of 0.44 to 0.82 $\mathrm{ms}^{-1}$. Supporting with these facts, we could conclude that emission rates from the trampling test areas were much greater than the zero trampling surface because of the

larger supplies of loose surface dust. It indicates the substantial effect of trampling (for dust loads) has been taken place on Mongolian temperate grassland, where is endured traditional animal husbandry for centuries. However, we are not able to give further insight at this point for increased dust contribution either from directly the availability of readily suspendable sediment or indirectly the process relationship between abrasive saltation by disturbance and dust emission, those are discussed in detail by (Macpherson et al., 2008; Baddock et al., 2011; Zhang et al., 2016).

It was demonstated that wind erosion and deposition processes forms uneven spatial distribution of dust supplements as driven by microclimatic, sedimentological, geochemical, surface patchiness and biological conditions (Gill, 1996). Likewise, we noticed larger standard deviations (Fig. 4b), those are resulted by scattered data points from their sample populations (see Fig. 4a; data points with box chart of $25^{th}$ and $75^{th}$ percentiles). Higher diversity of dust fluxes presents morphological disparity and sedimentological diversification presence in test areas. It may caused by as a result of the aeolian processes, tha dust emis-

sions are highly variable with space and between distinct landforms, even within individual landforms (Gill, 1996; Reynolds et al., 2007). Another reason it may related to that dust flux does not come to a similar saturation from a field site (Gillette and Passi, 1988). One of possible microscale disturbance by marmots creates spatially heterogeneous grasslands at a fine scale (Yoshihara et al., 2010). Moreover, it was emphasized that the livestock modified spatial heterogeneity at the landscape scale, whereas marmots modified spatial heterogeneity at the local scale (Yoshihara et al., 2010).

## 6.2   Effect strength of trampling

Very interestingly, we found that the variabilities in the mean ratio of $F_N$ and $F_{REF}$ is subject to changes in $u_*$ demonstrating positive relationships for all trampling test areas (Fig. 5a). As for an example, $F_{N_{250}}/F_{REF}$ ranged from 2.5 to 20 for $u_*$ values increased from 0.44 to 0.82 $\mathrm{ms}^{-1}$. It points that the effect strength of trampling get magnified with an increase in $u_*$. This means that dust emits much as stronger wind prevails at the trampled surfaces, like revealing 'hidden effect' of trampling. Some of the mean ratios below 1, particularly at low friction velocities for livestock trampling with $N_{201}$ and $N_{241}$ Headha$^{-1}$, supports the idea that (hidden) effect of trampling requires high $u_*$ to be revealed out. This $u_*$-magnified effect of trampling is a coincidence with the emissivity pattern of unlimited supply surfaces, discussed in Section 6.1. Our finding may hint the possible reason of huge amount of dust loading happenings when it originates from Mongolia (Laurent et al., 2006).

Beside of $u_*$, trampling density ($N$) also contribute the mean ratio variabilities to increase. It can be seen in the elevated mean ratios, those are increased from 10 times (at $N_{201}$) to 16 and 20 times (at $N_{241}$ and $N_{250}$, respectively), demonstrated for $u_*$ of 0.82 $\mathrm{ms}^{-1}$ (Fig. 5b). It is likely that increased $N$ results an elevated dust emission. However, the differences between $F_{N_{201}}/F_{REF}$ and $F_{N_{241}}/F_{REF}$ become smaller as the value in $u_*$ of 0.82 $\mathrm{ms}^{-1}$ shifts to moderate and low $u_*$ values (Fig. 5b). Relatedly, greater dust loading was manifested at the trampling density of $N_{250}$, and not, for $N_{241}$ and $N_{250}$ (Fig. 4b, denoted



by $*$). This indicates significantly higher dust loading occurs after a disturbance level has reached to $N_{250}$. Similar result of increased dust occurence with the disturbance level for cattle passes was presented (Baddock et al., 2011). Surprisingly, we observed that the disturbance level for the significant dust emission (comparably to $F_{REF}$) was lowered with an increase in wind force, $u_*$ (Fig. 4b). This suggests that both $u_*$ and $N$ has enormous combinational effect on dust emission from trampling test surface, and eventually, determines the effect strength of trampling. But to measuring the effect of trampling, in

an interactive (combinational) term of $u_*$ and $N$ is somewhat challenging.

In the livestock trampling function, almost 4 times greater of determinant power for $u_*$ than of for $N$ was presented. It points that the effect strength of trampling get magnified with an increase in $u_*$, and, therefore dust will emit much as stronger wind prevails at the trampled surfaces. This finding indicates that the effect of trampling can be seen or get into a play in emission when wind is strong.

**6.3  The applicability of $f_L(N, u_*)$ function for assessing anthropogenic dust**

We confirmed that eq.(8) has a feasible potential to assess dust emission fluxes, properly. Based on our eq.(8) (considering $F = F_{REF} + F_N$), anthropogenic dust is proportional to $u_*^8$. This is in a good agreement with dust fluxes proportional to $u_*^6$ for natural soil with abrasion and $u_*^{10}$ for fully unlimited supply soil (Zhang et al., 2016). Importantly, it implies that anthropogenic dust flux induced by trampling at a temperate grassland has a emission potent in between those have at natural

soil with abrasion and fully unlimited supply. However, the threshold friction velocity should be included further for a better prediction performance, precisely (Gillette and Passi, 1988; Belnap and Gillette, 1997). Moreover, dust emissions cannot be perfectly estimated (Shao, 2001; Uno et al., 2006) using only livestock density information due to the influence of many other environmental variables (Shinoda et al., 2011) such as soil aggregation Ishizuka et al. (2012), soil moisture (Fécan et al., 1998; Ishizuka et al., 2009), vegetation roughness (Kimura and Shinoda, 2010; Nandintsetseg and Shinoda, 2015) and surface

patchiness. Therefore, the direct use of our trampling function for assessing anthropogenic dust is limited to the temperate grassland. Implication should be given attention to the valid range for livestock density and friction velocity, $u_*$. The assessment should be on annual basis, but can be modified to the required time period if grazing route is known. The applicability of our dust flux formula to other areas beyond the study location could be accomplished with PI-SWERL tests over a wider geographic area.

As we mentioned, potential dust emission data obtained by PI-SWERL® , so that formula $F$, with eq.(8), will also provide a possible evaluation of potential anthropogenic dust. We calculated $N$ as a total livestock number, which needs to be resolved in the near future. We assumed that all types of livestock (small and large rumnitants) has the same effect on land surface trampling, irrespective of the size or distribution of the footprints. In addition, we made no distinction between the weights of

5  the different livestock species. However, the potential variability due to the difference in weights warrants further investigation. (Xu, 2014) tested the quantity of dust emitted from vehicles and found that it varied with the weight of the vehicles.



## 7   Conclusions

In this study, we aimed to reveal effect of trampling on dust emission processes, in a quantitave manner. For achieving study aim, we investigated effect of livestock trampling on dust emissions, developed a livestock trampling function for emission rate, and finally discussed the reliability of the trampling function and its applicability for assessing anthropogenic dust. Based on our results, we drawn major conclusions as:

- We found the substantial effect of trampling on dust emission. Dust emission resuspensive or successive behaviour with an increase in $u_*$, suggest that our sandy soil of temperate grassland is somewhat similar to a supply-limited surface with successive emission (Macpherson et al., 2008). In contrast, dust emission from trampling test areas presents that the disturbed, trampling surface is an unlimited-supply dust surface concerning its apparent increased emission rate with an increase in $u_*$ (Fig. 4), except the case of $0.64\mathrm{ms}^{-1}$ subtle declined to $0.64\mathrm{ms}^{-1}$. The effect of trampling on dust emissions was persistent throughout all friction velocities (varied from 0.44 to $0.82~\mathrm{ms}^{-1}$).

- Significantly higher dust loading was occurred after a certain disturbance level has reached by the trampling. But, significancy in the disturbance level was lowered with an increase in wind force, $u_*$. Our result suggests that both $u_*$ and $N$ has enormous combinational effect on dust emission from trampling test surface, and eventually, determines the effect strength of trampling. We successfully developed a livestock trampling function dependant on $u_*$ and $N$. In the livestock trampling function, almost 4 times greater of determinant power for $u_*$ than of for $N$ was presented ($f_L(N, u_*) \sim N^{1.1} u_*^4$). It points that the effect strength of trampling get magnified with an increase in $u_*$, and, therefore dust will emit much as stronger wind prevails at the trampled surfaces. This finding indicates that the effect of trampling can be seen or get into a play in emission when wind is strong.

- We confirmed that eq.(8) has a feasible potential to assess dust emission fluxes, properly. Based on our eq.(8) (considering $F = F_{REF} + F_N$), anthropogenic dust is proportional to $u_*^8$. This relationship, with eq.(8), provides a possible evaluation of potential anthropogenic dust. As we mentioned, our formula was developed based on potential dust emission data obtained by PI-SWERL® .

Finally, we recommend that a better management for livestock allocation coupled with strategies to prevent dust loads is needed, however, there are many uncertainties and assumptions to be improved in this study. The applicability of our result is feasible with a care to other areas beyond the study location.

## 8   Data availability

The underlying data can be found in the Appendix.





# Appendix A: Appendix A

## A1 Appendix A1

**Table 2.** Land surface and soil size characteristics in the study area

| | Measured dust emissions, $\mu\mathrm{gm}^2\mathrm{s}^{-1}$ | | | | | | | | | |
|---|---|---|---|---|---|---|---|---|---|---|
| | $F_{REF}$ | | | | | $F_{N_{250}}$ | | | | |
| $u_*$, ms$^{-1}$ | 0.44 | 0.54 | 0.64 | 0.73 | 0.82 | 0.44 | 0.54 | 0.64 | 0.73 | 0.82 |
| | 7.75 | 14.87 | 9.95 | 17.14 | 13.71 | 86.76 | 73.70 | 32.08 | 141.49 | 110.21 |
| | 7.58 | 16.11 | 9.60 | 25.77 | 32.44 | 5.92 | 125.49 | 87.57 | 677.31 | 928.18 |
| | 17.08 | 16.47 | 20.31 | 58.69 | 83.07 | 4.41 | 40.24 | 21.06 | 370.49 | 709.59 |
| | 6.88 | 16.62 | 9.22 | 28.64 | 23.24 | 7.18 | 120.81 | 72.79 | 571.93 | 1292.03 |
| | 12.21 | 15.00 | 7.32 | 39.68 | 32.53 | | | | | |
| | 19.59 | 26.07 | 11.60 | 79.69 | 37.85 | | | | | |
| | 2.07 | 4.51 | 2.54 | 23.12 | 38.46 | | | | | |
| Mean | 10.5 | 15.7 | 10.1 | 39.0 | 37.3 | 26.1 | 90.1 | 53.4 | 440.3 | 760.0 |
| SD | 6.2 | 6.3 | 5.4 | 22.6 | 22.0 | 40.5 | 40.6 | 31.9 | 236.4 | 495.3 |



## A2 Appendix A2

**Table 3.** Land surface and soil size characteristics in the study area

| | Measured dust emissions, $\mu gm^2 s^{-1}$ | | | | | | | | | |
| | $F_{N_{201}}$ | | | | | $F_{N_{241}}$ | | | | |
| $u_*$, ms$^{-1}$ | 0.44 | 0.54 | 0.64 | 0.73 | 0.82 | 0.44 | 0.54 | 0.64 | 0.73 | 0.82 |
|---|---|---|---|---|---|---|---|---|---|---|
| | 2.07 | 9.10 | 9.36 | 268.38 | 569.14 | 1.89 | 4.37 | 8.50 | 412.38 | 1469.22 |
| | 2.4 | 5.45 | 4.51 | 247.47 | 477.52 | 6.34 | 15.92 | 12.9 | 124.73 | 165.94 |
| | 1.97 | 5.06 | 4.19 | 215.55 | 433.93 | 72.99 | 14.31 | 7.07 | 24.52 | 23.24 |
| | 2.71 | 5.67 | 5.85 | 97.96 | 641.01 | 1.82 | 2.51 | 2.49 | 50.45 | 131.55 |
| | 2.97 | 9.64 | 7.49 | 229.35 | 648.04 | 5.67 | 20.25 | 13.4 | 153.83 | 143.61 |
| | 4.09 | 11.62 | 10.44 | 222.24 | 452.61 | 1.99 | 8.14 | 6.04 | 42.57 | 55.1 |
| | 2.94 | 5.68 | 6.42 | 146.43 | 223.69 | 1.97 | 7.37 | 7.06 | 191.46 | 899.14 |
| | 3.21 | 14.28 | 12.41 | 160.77 | 365.88 | 4.01 | 29.91 | 19.13 | 246.65 | 551.89 |
| | 2.75 | 6.13 | 5.15 | 217.09 | 533.54 | 6.83 | 11.98 | 7.16 | 26.54 | 31.83 |
| | 2.73 | 5.61 | 4.85 | 85.18 | 177.24 | 2.26 | 8.52 | 6.12 | 137.08 | 596.04 |
| | 2.37 | 8.57 | 12.44 | 276.89 | 450.74 | 3.31 | 18.58 | 33.96 | 300.54 | 923.68 |
| | 3.41 | 9.2 | 7.78 | 170.03 | 728.82 | 5.25 | 17.48 | 15.43 | 213.66 | 1042.76 |
| | 3.81 | 8.02 | 17.81 | 308.82 | 712.61 | 4.08 | 27.73 | 29.94 | 432.75 | 1472.39 |
| | 3.76 | 8.9 | 11.64 | 259.24 | 597.8 | 8.05 | 44.2 | 42.09 | 433.13 | 1592.87 |
| | 31.04 | 48.44 | 15.28 | 244.37 | 354.01 | 30.71 | 53.14 | 26.27 | 143.26 | 180.92 |
| | 5.52 | 13.13 | 9.85 | 145.49 | 210.6 | 3.56 | 12.87 | 6.02 | 47.52 | 107.3 |
| | 8.4 | 14.43 | 15.37 | 261.1 | 661.45 | 5.7 | 10.29 | 5.31 | 16.61 | 10.14 |
| | 18.14 | 36.23 | 20 | 79.97 | 93.23 | 3.62 | 39.83 | 33.47 | 369.67 | 365.11 |
| | 3.56 | 8.09 | 7.07 | 48.61 | 46.17 | 4.98 | 15.48 | 9.29 | 62.39 | 56.63 |
| | 6.06 | 30.54 | 57.19 | 330.62 | 253.18 | 1.98 | 15.89 | 14.71 | 319.52 | 1221.66 |
| | 10.36 | 57.89 | 6.34 | 109.65 | 81.43 | 8.08 | 27.42 | 15.96 | 262.56 | 1604.88 |
| | 4.69 | 6.57 | 3.77 | 18.18 | 36.43 | | | | | |
| | 2.15 | 2.52 | 2.71 | 90.26 | 136.25 | | | | | |
| | 1.58 | 2.94 | 2.34 | 90.37 | 197.62 | | | | | |
| | 1.2 | 3.64 | 2.12 | 33.32 | 115.36 | | | | | |
| Mean | 5.4 | 13.5 | 10.5 | 174.3 | 367.9 | 8.8 | 19.3 | 15.3 | 191 | 602.2 |
| SD | 6.4 | 14.3 | 10.9 | 91 | 228.4 | 15.9 | 13.4 | 11.3 | 143.2 | 590.6 |





## A3  Appendix A3

It is necessary to provide the uncertainty of dust flux formula, $\delta F$, ( since Eq.(1) is derived based on the mean values $F_N$ for $N$. We estimated uncertainties of the calculated dust flux for each $u_*$, thus the uncertainty will be converted to Eq.(A1) by substituting Eq.3; and, considering $u_*$ as a constant for a given friction velocity:

$\delta F = |cu_*^4|\, \delta(1 + f_L(N, u_*))$   $\underrightarrow{\text{substitution of } F_N/F_{REF}}$   will yield   $\delta F = |cu_*^4|\, \delta(F/F_{REF})$   (A1)

If we apply uncertainty analysis for multiplication(**??**Taylor, 1997) into Eq.(A1), we will obtain Eq.A2,

$$\delta F = |cu_*^4|\,|\frac{F}{F_{REF}}|\sqrt{\left(\frac{\delta F}{F}\right)^2 + \left(\frac{\delta F_{REF}}{F_{REF}}\right)^2}$$   (A2)

where $\delta F$ and $\delta F_{REF}$ are uncertainties of $F$ and $F_{REF}$, respectively. Hereafter, we estimated $\delta F$ and $\delta F_{REF}$ using the basic statistical methodology for calculating uncertainty as a standard error of the mean value ((Taylor, 1997; Coleman and

Steele, 2009). It has been proven by Taylor (1997) that the standard error of the mean value for a certain population is defined as that a standard deviation of a given population ($SD$) is divided by the square root of its population number ($\sqrt{n}$). Thus, we calculated the uncertainty in the means of F as $\delta F = \dfrac{SD}{\sqrt{n}}$. As for a small dataset (Taylor (1997); Coleman and Steele (2009)), we calculated uncertainty in the means of $F_{REF}$ as $\delta F_{REF} = \dfrac{F_{REF,max} - F_{REF,min}}{2\sqrt{n}}$ using the maximum ($F_{REF,max}$) and minimum ($F_{REF,min}$) values among the dataset for the non-trampled surfaces. For defining $\delta F_{REF}$, first we

solved $\sqrt{\left(\frac{\delta F}{F}\right)^2 + \left(\frac{\delta F_{REF}}{F_{REF}}\right)^2}$ as 0.41, 0.31, 0.37, 0.32, and 0.37 for each $u_*$ of 0.44, 0.54, 0.64, 0.73, and 0.82 ms$^{-1}$. For

general use, we adopted the mean of the solved $\sqrt{\left(\frac{\delta F}{F}\right)^2 + \left(\frac{\delta F_{REF}}{F_{REF}}\right)^2}$ values, thus $\delta F$ is defined as in Eq.A3

$\delta F = 0.36 F$   (A3)

We can interpret the uncertainty of dust flux is 0.36 times as much as the calculated dust flux (Eq. 3S).





*Acknowledgements.*  The authors gratefully thank Dr. James King, Universite de Montréal, Canada for his support during the PI-SWERL® experiment in the field. We also specially thank Dr. Battur Gompil, National University of Mongolia for his useful advice on Cobb-Douglass formula for the manuscript. This study was supported by the Grant-in-Aid for Scientific Research (S) from JSPS 'Integrating Dryland Disaster Science' (No.25220201) at Nagoya University and 'Advanced Research Grant 2017' at National University of Mongolia, Mongolia.




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
