# Peer review of "An Anthropogenic Dust Emissions due to Livestock Trampling in a Mongolian Temperate Grassland"

_Atmospheric Chemistry and Physics, 2017_

## Referee Comment (RC1) · Anonymous Referee #1 · 17 Mar 2017

This paper describes an experimental study on the effect of livestock trampling on dust emission using a mini wind tunnel. The subject is interesting from the anthropogenic dust point of view. However, the results presented in the paper seems not very reasonable (in my opinion) about the dust emission mechanism. Also, it does not provide sufficient information for estimating the impact of livestock trampling on the regional dust emission amount. From the dust emission mechanism point of view, there should be more detailed descriptions about the change in the physical condition of grassland surface by trampling. The ratio of the flux F/F(free) is considered in this analysis. However, the reason for taking the ratio is not explained. Physical meaning is not clear. (In addition, the data for F(free) contains big errors as can be seen in Appendix.) Also,

the reason for describing the ratio by 1+f is not clear. It seems the function form to describe the experimental results is rather arbitrary. In Eq. (8), the major term that represents the effect of trampling is proportional to the 8th power of $u^*$. Is it reasonable? A physically reasonable function form should be used. An important subject would be estimating the contribution of dust emission from trampled grassland. From this point of view, it would be better to discuss the dust emission flux in (trampled) grassland in comparison with that in arid and semi-arid regions.

---

## Referee Comment (RC2) · Anonymous Referee #2 · 27 Mar 2017

This paper examines the effect of livestock trampling on the potential for dust emissions from Mongolian Steppe grass landscape. The basic approach is to use wind tunnel measurements (PI-SWERL) to measure windblown emissions on background (untrampled) surfaces and subsequently trampled surfaces. Two types of measurements are conducted. The first is in a controlled pen where livestock grazing was not allowed for some time. This area was apparently used to obtain background (undisturbed/untrampled) levels of dust emissions. The area was then grazed by livestock (estimated density of 250 head per hectare) and re-measured for emissions. In separate tests, a transect of wind tunnel measurements was conducted in 2009 and again in 2010 through an area of actual livestock grazing with estimated densities of 201 and

241 head per hectare. The results are analyzed for dust emissions increases due to livestock grazing at different values of friction velocity.

Overall, this information is very valuable to the understanding of wind erosion, especially in the Mongolian Steppe. These kind of data are lacking from the literature and can be very helpful for estimating anthropogenic impacts on the dust problems within Asia. I have two major concerns with the manuscript. The first is that it can be rather difficult to follow the discussion because the presentation is not always clear. I found myself having to guess the meaning of a lot of sentences and infer information where I would have liked to know for sure that the authors were conveying a specific point. The second is that while the data collected are quite valuable, the variation in conditions is rather limited and it is not clear that the extensive development of functional relationships is warranted or justified. These include somewhat limited temperature, relative humidity, and antecedent soil moisture information. They also include a limited range of livestock trampling which span from 201 to 250 head per hectare and are based on necessarily coarse estimates since it is difficult to know precisely how many sheep trampled over what area during what period of time.

In my view, both of these concerns can be addressed by substantially shortening the manuscript and sticking strictly to the main findings. Organizationally, I suggest greatly shortening the introductory and background information to the minimum necessary to convey the importance of livestock grazing to Asian dust. Technically, I recommend that rather than fitting a function of u* and N, it would be better to simply provide an enhancement factor of emission due to livestock trampling at different values of u* and stating that the information applies to N $\sim$ 250 head per hectare. The exact representation is up to the authors, but one option would be a curve that has u* on the x-axis and enhanced emissions on the y-axis (FN/FREF in the notation of the manuscript, similar to what is now depicted in Figure 5a and b). This would also help show that there is apparently a u* below which there appears to be little difference in emissions between trampled and untrampled steppe soil. Perhaps the discussion can then focus on the

limited nature of the study and where additional information would be most helpful in future work.

---

## Author Comment (AC1) · 22 Jun 2017

[answers,12pt]exam

**Response to Referee #1 :**
We appreciate the Editor and Referee # 1 for their valuable and constructive comments for this manuscript, which greatly assist in improving the quality of the original manuscript. We have carefully checked and revised the whole manuscript according to the Referee #1 's comments. Please find a point-by-point reply to the issues as follows. And we have also uploaded the file of 'acp-2017-94-revised.pdf'.

[Figure]

**Comments raised by Referee # 1**

This paper describes an experimental study on the effect of livestock trampling on dust emission using a mini wind tunnel. The subject is interesting from the anthropogenic dust point of view. However, the results presented in the paper seems not very reasonable (in my opinion) about the dust emission mechanism. Also, it does not provide sufficient information for estimating the impact of livestock trampling on the regional dust emission amount. From the dust emission mechanism point of view, there should be more detailed descriptions about the change in the physical condition of grassland surface by trampling. The ratio of the flux F/F(free) is considered in this analysis. However, the reason for taking the ratio is not explained. Physical meaning is not clear. (In addition, the data for F(free) contains big errors as can be seen in Appendix.) Also, the reason for describing the ratio by 1+f is not clear. It seems the function form to describe the experimental results is rather arbitrary. In Eq. (8), the major term that represents the effect of trampling is proportional to the 8th power of u*. Is it reasonable? A physically reasonable function form should be used.

An important subject would be estimating the contribution of dust emission from trampled grassland. From this point of view, it would be better to discuss the dust emission flux in (trampled) grassland in comparison with that in arid and semi-arid regions.

>**Response:** We greatly thank Anonymous Referee # 1 for quite helpful and constructive comments given to our manuscript (MS). We highly appreciate your contribution for the improvements of our manuscript with well-judged comments.
>
>We have revised our manuscript according to the suggestions and comments raised by the Referee # 1. We strongly believe that the Referee # 1 will satisfy with the revised version of the manuscript. The major revisions are included:
>
>   1. We agree to the Referee's point given on the livestock trampling function ($f_L$). The equation of the function was derivative by statistical fitting, not based on dust emission mechanism. We have revised our manuscript according to the Referee's point. The livestock trampling function (fL) is no longer in the central focus of this MS. Instead, MS focuses on the dust emission at trampled grassland ($F_N$) in a comparison to dust emission from natural grassland ($F_{REF}$), as it suggested by the Referee # 1 in the concluding paragraph on the review letter.

2. We have used the ratio of $F_N/F_{REF}$ to differentiate between the different dust strengths, and by it, we scaled factor of anthropogenic dust emission due to livestock trampling. We have included the explanation for scale factor in the 'Data analysis section'.

3. Since we have excluded $f_L$ function, $1 + f_L$ term is no longer in the revised manuscript.

**Point by point response**

1. This paper describes an experimental study on the effect of livestock trampling on dust emission using a mini wind tunnel. The subject is interesting from the anthropogenic dust point of view.

   **Response:** Thank you. This comment is very encouraging for us.

2. However, the results presented in the paper seems not very reasonable (in my opinion) about the dust emission mechanism. Also, it does not provide sufficient information for estimating the impact of livestock trampling on the regional dust emission amount.

   **Response:** We appreciate Referee's concern. We considered the point, carefully. It is crucial to estimate the impact of livestock trampling

on the regional dust emission amount. However, our limited dataset restricts us to provide sufficient information on a regional level. With this aim, we tried to develop the livestock trampling function as an applicable simple formula to estimate dust emissions from trampled Mongolian steppe (which occupy approx. 30% of its territory). The function is simple; depends on $u_*$ and $N$. These kind of data can be downloadable on annual basis. Moreover, it will be easier to account the livestock number, than defining changes in soil physical conditions (Referee # 1 also mentioned, we should also check how soil physical condition changes). In practical application, a such derivation would be handful. However, the formula is fitted statistically; which fails to express its physical meaning. We completely agree to the Referee's point that the formula does not reflect dust emission physics. Therefore, we revised content of the MS into "scale factor" as a ratio of $F_N$ to $F_{REF}$.

```
[Data analysis section: Page 12, Line 4-14]
```

And, we emphasized the necessity on larger dataset covering spatial variations and soil textures, as well as atmospheric conditions for a broad usage of the scale factor.

```
[Page 19, Line 19-20; Page 20, Line 23-24]
```

3. From the dust emission mechanism point of view, there should be more detailed descriptions about the change in the physical condition of grassland surface by trampling.

   **Response:** Thank you very much for this comment. As the Referee mentioned, it would be a perfect illustration if we could link trampling effect on soil surface; and then connect it to the dust emission. This

type of work is needed for a better understanding and a well described anthropogenic dust emission mechanism. The referee is right to point it out, yet our research focus is the dust emission strength changes due to trampling. Our research is no longer focused on the dust mechanism.

It is true that the soil physical change by trampling is the study topic we should push forward. Importantly, in our study, the factual effect of trampling on dust emission is revealed; this foremost step will direct to the advanced study. We strongly believe that the Referee will agree to it. We have already included the brief descriptions of soil physics changes resulted from mechanical disturbance by livestock trampling. We believe this type of information is satisfactory in the content of our MS.

    [Page 2, Line 30-31; Page 20, Line 2-5]

4. The ratio of the flux F/F(free) is considered in this analysis. However, the reason for taking the ratio is not explained. Physical meaning is not clear. (In addition, the data for F(free) contains big errors as can be seen in Appendix.)

    **Response:** Thank you for pointing it out. Now, we have included the meaning of the ratio; and the reason. The ratio of $F_N/F_{REF}$ is presented to differentiate between the different dust strengths. The ratio not only differentiates dust strengths but also it expresses / measures a scale factor of dust emission enhanced by trampling.

    [Page 12, Line 7-13]

    As for obtained data, we too were disappointed by the big errors (in Appendix). Later, we have been convinced that it is the correct. On
the field, we had recognized heterogenous sediment surface bye eye observation. The possible reason for heterogeneous sediment distribution was demonstrated by the other research findings. We separately added a detailed explanation on the relatively big standard deviations on measured dust emissions on mean dust emission sub section of 'Methods and Materials section'.

[Page 11, Line 9-20]

5. Also, the reason for describing the ratio by 1+f is not clear. It seems the function form to describe the experimental results is rather arbitrary.

**Response:** Right. We completely agree with the comment. Yes, the function is fitted, statistically. It will give some uncertainty for estimating impact of trampling effect on dust emissions; particularly for the areas beyond our study site. Therefore, we recommend that to test the same for more broad illustration. This kind of result with updated data will be useful; and connecting and combining the results from other sites will surely give us a big picture.

Since, we focused on the scale factor of dust emission due to trampling (dust emission ratio from trampled to natural grassland) by removing fl function; at this time we have also excluded our $1 + f_L$ result from MS.

[Page 10, Line 13-19; Page 17]

6. In Eq. (8), the major term that represents the effect of trampling is proportional to the 8th power of u*. Is it reasonable? A physically reasonable function form should be used.

**Response:** Yes, somehow. We are not certain; this result should confirmed by other results. One of the confirmation was the recent finding

by Zhang et al. 2016; showed that the dust emission dependent on $u*^4 - u*^{10}$ varied by the renewal process. This renewal process (disturbance) is kind of artificial disturbance. Our result lies in the range, which was discussed in the MS. However, as the Reviewer mentioned we should aim for the physically based formula. Again, now we focused on the contribution of the dust from the trampled grassland by removing fl function; therefore the related changes excluded such a discussion.

```
[Page 19, Line 12-14]
```

7. An important subject would be estimating the contribution of dust emission from trampled grassland. From this point of view, it would be better to discuss the dust emission flux in (trampled) grassland in comparison with that in arid and semi-arid regions.

> **Response:** Thank you very much. This comment gave us a clear directional insight how to revise our manuscript. We totally agree to the Referee's concluding comment. In the revised manuscript, a comparison between dust emissions from trampled surface and non-trampled surface is the main research point. In the previous version of the manuscript, the major point was cluttered by the livestock trampling function. We have revised our manuscript according to the Referee's comments.

**Supplement:**

[revised manuscript text omitted]

---

## Author Comment (AC2) · 22 Jun 2017

[answers,12pt]exam

**Response to Referee #2 :**
We appreciate the Editor and Referee # 2 for their valuable and constructive comments for this manuscript, which greatly assist in improving the quality of the original manuscript. We have carefully checked and revised the whole manuscript according to the Referee #2's comments. Please find a point-by-point reply to the issues as follows. And we have also uploaded the revised manuscript file of 'acp-2017-94-revised.pdf'.

[Figure]

**Comments raised by Referee # 2**
This paper examines the effect of livestock trampling on the potential for dust emissions from Mongolian Steppe grass landscape. The basic approach is to use wind tunnel measurements (PI-SWERL) to measure windblown emissions on background (untrampled) surfaces and subsequently trampled surfaces. Two types of measurements are conducted. The first is in a controlled pen where livestock grazing was not allowed for some time. This area was apparently used to obtain background (undisturbed/ untrampled) levels of dust emissions. The area was then grazed by livestock (estimated density of 250 head per hectare) and re-measured for emissions. In separate tests, a transect of wind tunnel measurements was conducted in 2009 and again in 2010 through an area of actual livestock grazing with estimated densities of 201 and 241 head per hectare. The results are analyzed for dust emissions increases due to livestock grazing at different values of friction velocity.

Overall, this information is very valuable to the understanding of wind erosion, especially in the Mongolian Steppe. These kind of data are lacking from the literature and can be very helpful for estimating anthropogenic impacts on the dust problems within Asia.

I have two major concerns with the manuscript. The first is that it can be rather difficult to follow the discussion because the presentation is not always clear. I found myself having to guess the meaning of a lot of sentences and infer information where I would have liked to know for sure that the authors were conveying a specific point. The second is that while the data collected are quite valuable, the variation in conditions is rather limited and it is not clear that the extensive development of functional relationships is warranted or justified. These include somewhat limited temperature, relative humidity, and antecedent soil moisture information. They also include a limited range of livestock trampling which span from 201 to 250 head per hectare and are based on necessarily coarse estimates since it is difficult to know precisely how many sheep trampled over what area during what period of time.

In my view, both of these concerns can be addressed by substantially shortening the manuscript and sticking strictly to the main findings. Organizationally, I suggest greatly shortening the introductory and background information to the minimum necessary to convey the importance of livestock grazing to Asian dust. Technically, I recommend that rather than fitting a function of $u_*$ and $N$, it would be better to simply provide an enhancement factor of emission due to livestock trampling at different values of $u_*$ and stating that the information applies to N 250 head per hectare. The exact representation is up to the authors, but one option would be a curve that has $u_*$ on the x-axis and enhanced emissions on the y-axis ($F_N/F_{REF}$ in the notation of the manuscript, similar to what is now depicted in Figure 5a and b). This would also help show that there is apparently a $u_*$ below which there appears to be little difference in emissions between trampled and untrampled steppe soil. Perhaps the discussion can then focus on the limited nature of the study and where additional information would be most helpful in future work.

**Response:** We completely agree to the Referee′s comments given on our manuscript (MS); and by it, we have substantially improved quality of our MS. We strongly believe that the Referee # 2 will satisfy with the revised version of the manuscript. We greatly appreciate your contribution for the improvements of our manuscript giving us well-instructed comments. The major revisions are included:

1. The livestock trampling function ($f_L$) is excluded from our MS. Our revised MS focuses on the scale factor of dust emission due to trampling as the Referee # 2 suggested.

2. We have shortened 'Introduction section'. We have added a tabular chart of a scale factor for a better illustration, and revised texts in 'Discussion section' for a clarity.

3. Since we have sticked to the main results, the related and associated parts (title, abstract, result, discussion) are justified and corrected as well.

**Point by point response**

1. This paper examines the effect of livestock trampling on the potential for dust emissions from Mongolian Steppe grass landscape. The basic approach is to use wind tunnel measurements (PI-SWERL) to measure windblown emissions on background (untrampled) surfaces and subsequently trampled surfaces. Two types of measurements are conducted. The first is in a controlled pen where livestock grazing was not allowed for some time. This area was apparently used to obtain background (undisturbed/ untrampled) levels of dust emissions. The area was then grazed by livestock (estimated density of 250 head per hectare) and re-measured for emissions. In separate tests, a transect of wind tunnel measurements was conducted in 2009 and again in 2010 through an area of actual livestock grazing with estimated densities of 201 and 241 head per hectare. The results are analyzed for dust emissions increases due to livestock grazing at different values of friction velocity.

Overall, this information is very valuable to the understanding of wind erosion, especially in the Mongolian Steppe. These kind of data are lacking from the literature and can be very helpful for estimating anthropogenic impacts on the dust problems within Asia.

> **Response:** Thank you very much for your careful reading of our manuscript. We greatly appreciate your positive comment and given a value on our work.

2. I have two major concerns with the manuscript. *The first* is that it can be rather difficult to follow the discussion because the presentation is not always clear. I

found myself having to guess the meaning of a lot of sentences and infer information where I would have liked to know for sure that the authors were conveying a specific point. *The second* is that while the data collected are quite valuable, the variation in conditions is rather limited and it is not clear that the extensive development of functional relationships is warranted or justified. These include somewhat limited temperature, relative humidity, and antecedent soil moisture information. They also include a limited range of livestock trampling which span from 201 to 250 head per hectare and are based on necessarily coarse estimates since it is difficult to know precisely how many sheep trampled over what area during what period of time.

> **Response:** We totally agree to the two major points raised by the Referee # 2.
> - Firstly, we have revised 'Discussion section' according to the Referee's point by adding a tabular chart for scale factor (Fig.7) and editing text.
>
> `[Page 19, Figure 7; Page 17]`
>
>
> - Secondly, our data was obtained in the limited condition. Yes, the further study is needed to investigate and develop a such functional relationship as Referee # 2 mentioned. Livestock trampling time span is on annual. We have clarified it in the manuscript.
>
> `[Page 8, Line 15]`

3. In my view, both of these concerns can be addressed by substantially shortening the manuscript and sticking strictly to the main findings.

   Organizationally, I suggest greatly shortening the introductory and background

information to the minimum necessary to convey the importance of livestock grazing to Asian dust.

Technically, I recommend that rather than fitting a function of $u_*$ and $N$, it would be better to simply provide an enhancement factor of emission due to livestock trampling at different values of $u_*$ and stating that the information applies to N 250 head per hectare. The exact representation is up to the authors, but one option would be a curve that has $u_*$ on the x-axis and enhanced emissions on the y-axis ($F_N/F_{REF}$ in the notation of the manuscript, similar to what is now depicted in Figure 5a and b). This would also help show that there is apparently a $u_*$ below which there appears to be little difference in emissions between trampled and untrampled steppe soil.

**Response:** We highly appreciate comments raised by Referee # 2. We have revised our manuscript according to the Referee's point.
- We have shortened 'Introduction section' as recommended by Referee # 2.

`[Page 2-3]`

- The livestock trampling function (fL) is no longer in the central focus of this MS. Instead, MS focuses on the (enhancement) scale factor of emission due to livestock trampling as the Referee # 2 suggested. We have separated Figure 5a and b into two figures (Figure 5 and Figure 6) to present the $u_*$-magnified and $N$-elevated scale factor. For a better presentation (and more clarified 'Discussion section'), we have added a tabular chart of scale factor in Figure 7.

`[Page 15, Figure 5; Page 17, Figure 6; Page 19, Figure 7]`

4. Perhaps the discussion can then focus on the limited nature of the study and where additional information would be most helpful in future work.

   **Response:** Thank you for pointing it out. We obtained dust data on the limited nature; so that our result represents the limited condition. Yes, we agree to the comment. It is important to mention our study limitations and mention additional information those are needed to be explored for future research work. We have discussed the limited nature of the study and future work.

   `[Page 19, Line 17-24; Page 20, Line 1-6; 12]`

**Supplement:**

[revised manuscript text omitted]